# Solar Photocatalytic Membranes: An Experimental and Artificial Neural Network Modeling Approach for Niflumic Acid Degradation

**DOI:** 10.3390/membranes12090849

**Published:** 2022-08-30

**Authors:** Lamine Aoudjit, Hugo Salazar, Djamila Zioui, Aicha Sebti, Pedro Manuel Martins, Senentxu Lanceros-Méndez

**Affiliations:** 1Unité de Développement des Équipements Solaires, UDES/Centre de Développement des Energies Renouvelables, CDER, Bou Ismail 42415, Algeria; 2Physics Centre of Minho and Porto Universities (CF-UM-UP), University of Minho, Campus de Gualtar, 4710-057 Braga, Portugal; 3LaPMET—Laboratory of Physics for Materials and Emergent Technologies, University of Minho, Campus de Gualtar, 4710-057 Braga, Portugal; 4Centre/Department of Chemistry, University of Minho, Campus de Gualtar, 4710-057 Braga, Portugal; 5Centre of Molecular and Environmental Biology, University of Minho, Campus de Gualtar, 4710-057 Braga, Portugal; 6Institute of Science and Innovation on Bio-Sustainability (IB-S), University of Minho, 4710-057 Braga, Portugal; 7BCMaterials, Basque Centre for Materials, Applications and Nanostructures, UPV/EHU Science Park, 48940 Leioa, Spain; 8IKERBASQUE, Basque Foundation for Science, 48013 Bilbao, Spain

**Keywords:** artificial neural network, modelling, nanocomposite membrane, niflumic acid, photocatalysis, photocatalytic membrane reactor

## Abstract

The presence of contaminants of emerging concern (CEC), such as pharmaceuticals, in water sources is one of the main concerns nowadays due to their hazardous properties causing severe effects on human health and ecosystem biodiversity. Niflumic acid (NFA) is a widely used anti-inflammatory drug, and it is known for its non-biodegradability and resistance to chemical and biological degradation processes. In this work, a 10 wt.% TiO_2_/PVDF–TrFE nanocomposite membrane (NCM) was prepared by the solvent casting technique, fully characterized, and implemented on an up-scaled photocatalytic membrane reactor (PMR). The photocatalytic activity of the NCM was evaluated on NFA degradation under different experimental conditions, including NFA concentration, pH of the media, irradiation time and intensity. The NCM demonstrated a remarkable photocatalytic efficiency on NFA degradation, as efficiency of 91% was achieved after 6 h under solar irradiation at neutral pH. The NCM proved effective in long-term use, with maximum efficiency losses of 7%. An artificial neural network (ANN) model was designed to model NFA’s photocatalytic degradation behavior, demonstrating a good agreement between experimental and predicted data, with an R^2^ of 0.98. The relative significance of each experimental condition was evaluated, and the irradiation time proved to be the most significant parameter affecting the NFA degradation efficiency. The designed ANN model provides a reliable framework l for modeling the photocatalytic activity of TiO_2_/PVDF-TrFE and related NCM.

## 1. Introduction

Water contamination is an alarming global concern nowadays due to its direct influence on human health and ecosystem biodiversity [1]. Due to their extensive use and uncontrolled release, pharmaceuticals are among the most critical contaminants of emerging concern (CEC) for water. Their presence in aqueous effluents is detected at concentrations from ng/L to mg/L, representing a detrimental impact on aquatic ecosystems and humans [2].

Niflumic acid (NFA), which is extensively used as an analgesic, anti-arthritic, and anti-rheumatic, has been gaining special attention among the non-steroidal anti-inflammatory drugs, mainly due to its high solubility in water and non-biodegradability, making it frequently detected in wastewater treatment plants [3]. The extended presence of NFA might be responsible for irreparable damage to the environment and human health, such as physiological disorders, skeletal and dental fluorosis, and kidney damage, as well as in plants, inhibiting germination, leading to ultra-structural malformations, and reducing photosynthetic capacities and productivity [4].

The extensive presence of NFA in water bodies is caused by its ineffective removal from water effluents by traditional wastewater treatment technologies, such as ozonation [5], gravity separation [6], ultrasonic separation [7], adsorption [8], and coagulation/flocculation [9], among others. Further, these techniques present some drawbacks, such as implementation and operation costs and the generation of secondary pollutants [10]. To surpass the limitations associated with these conventional techniques, strong efforts are being carried out to develop more advanced approaches that can ensure effective removal of this contaminant from wastewater. In this scope, advanced oxidation processes and, in particular, photocatalysis, arise as interesting approaches for removing CECs [11].

Photocatalysis allows a complete degradation of CECs into smaller and harmless compounds through oxidation reactions with free radicals generated by a photocatalyst [12]. Titanium dioxide (TiO_2_) is one of the most used photocatalysts, as it presents high photocatalytic activity, chemical and thermal stability, superhydrophilicity, large surface area, reduced toxicity, and low cost [13]. Despite these properties, applying TiO_2_ nanoparticles (NPs) in wastewater treatment implies further treatments for their recovery and reuse. To overcome this limitation, its immobilization into polymeric substrates, whose porosity and pore size can be controlled, is an attractive alternative [14].

Membrane-based technologies, in particular, membranes based on poly(vinylidene fluoride) (PVDF) and its copolymers are of high interest for wastewater treatment due to their outstanding properties such as mechanical, chemical, thermal, and UV stability [15]. Membranes based on PVDF have been investigated for the removal of different contaminants from water, such as heavy metals [15], natural organic matter [16], proteins [17], volatile organic compounds [18], desalination [19], and organic dyes [20]. Among the different PVDF co-polymers, poly(vinylidene fluoride-trifluoroethylene) (PVDF-TrFE) presents high UV, bio and chemical stability, biocompatibility, and allows the production of membranes with a controlled porosity and pore size [21]. The incorporation of TiO_2_ into polymeric membranes has already proven to be suitable for wastewater treatment, showing remarkable photocatalytic activity in the degradation of tartrazine, with sunlight photocatalytic degradation of 78% over 5 h [13].

The implementation of a nanocomposite membrane (NCM) containing TiO_2_ and PVDF-TrFE in a photocatalytic membrane reactor (PMR) improves the process efficiency and stability, since it allows the reutilization of the photocatalyst, controls the contact time between photocatalyst and contaminant, and allows a continuous process, making PMR an efficient, environmentally friendly and low-cost wastewater treatment technology [22].

Further, the predictive evaluation of the contaminant degradation efficiency is also of great interest since it allows the formulation of models capable of forecasting the performance when it comes to the degradation of a specific contaminants, and properly tune materials and methods to optimize the functional application or the photocatalytic membranes [23,24].

In recent years, artificial neural network (ANN) has become a powerful tool capable of resolving complex problems of classification, pattern recognition, clustering, and prediction [25]. It has been widely applied in different fields, including the mining industry [26], civil engineering [27], geotechnical and geoengineering [28], disaster risk assessment [29] and environmental engineering [30], among others.

ANNs are empiric models inspired by the structure and functioning of biological neural networks, based on the concept that a highly interconnected system composed of simple processing elements, called neurons or nodes, empowers the learning of complex non-linear relationships between input and output variables [31]. In the architecture of an ANN, neurons are associated by weights in a parallel structure consisting of: (1) an input layer, which contains neurons as independent variables of the problem; (2) hidden layers, which will receive information from the input layer and implement calculations following training algorithms; (3) an output layer, where the prediction is displayed [32]. Neurons, or nodes, are the primary units in each layer and are connected between all the layers by algorithms and weights. Therefore, the neurons are passive since they are only used to receive the data patterns from the external source and transmit them to the subsequent hidden layer. Hidden layers are the key to learning the data pattern and introducing the nonlinearity into the network [33,34].

In the context of this work, an ANN was used to develop a predictive model for the performance of the TiO_2_/PVDF-TrFE membrane in NFA photocatalytic degradation, demonstrating the suitability of this material to be implemented in an up-scaled photocatalytic membrane reactor as an efficient approach for the degradation of NFA under solar irradiation and environmental conditions.

## 2. Experimental

### 2.1. Chemicals

Poly(vinylidene fluoride-trifluoroethylene) (PVDF-TrFE), with a molecular weight of 350,000 g/mol, was purchased from Solvay (Brussels, Belgium). TiO_2_ (P25-AEROXIDE) NPs were acquired from Evonik Industries AG (Essen, Germany). N, N-dimethylformamide (DMF) was purchased from Merck (St. Louis, MO, USA). Niflumic acid, C_13_H_9_F_3_N_2_O_2_, with a molecular weight of 282.22 g/mol and a maximum absorption wavelength of 286 nm, was purchased from Glenmark Pharmaceutical (Mumbai, India). Sodium hydroxide (NaOH) and hydrochloric acid (HCl) were obtained from HACH Company (Loveland, CO, USA).

### 2.2. Production of TiO_2_/PVDF-TrFE Nanocomposite Membranes

The 10 wt.% TiO_2_/PVDF-TrFE nanocomposite membranes were prepared by the solvent casting technique, according to previous works [20,35]. Briefly, a defined amount (10 wt.%) of TiO_2_ NPs were dispersed in DMF by ultrasonication for 3 h, to obtain a homogeneous dispersion. The amount of TiO_2_ NPs used was defined concerning previous works [20] to optimize the performance of the membranes while preserving their mechanical properties and preventing the detachment of the NP from the polymer matrix. Afterwards, PVDF-TrFE was added to the dispersion, achieving a polymer/solvent ratio of 10:90 *v*/*v*, and the solution was magnetically stirred until complete dissolution of the polymer. Finally, the solution was poured into a glass support at room temperature for slow solvent evaporation. NCMs with the same dimensions of the photoreactor surface (38 cm × 12 cm × 600 µm) were precisely cut out after the complete evaporation of the solvent.

### 2.3. Nanocomposite Membranes Characterization

The crystalline structure of the TiO_2_ NPs was evaluated by X-ray diffraction (XRD), using a Bruker D8 Discover diffractometer with an incident Cu K_α_ radiation (40 kV and 30 mA). Scanning Electron Microscopy (SEM) was performed to access the morphology and microstructure of NCM, using a Thermo Fisher Quanta 650 SEM apparatus with an accelerating voltage of 10 kV. Before SEM analysis, the NCM was coated with a thick gold layer by magnetron sputtering. Fourier transformed infrared spectroscopy (FTIR), in attenuated total reflectance (ATR) mode, was employed using a Jasco FT/IR-4100 apparatus. Analyses were performed in the spectral range from 650 to 4000 cm^−1^, using 64 scans with a resolution of 4 cm^−1^. Differential scanning calorimetry (DSC) analysis was performed using a Mettler Toledo DSC 822e apparatus in the temperature range of 25–200 °C, a heating rate of 10 °C/min, and flowing nitrogen atmosphere. The wetting characteristics of the NCMs were evaluated through contact angle measurements using a Physics SCA20 microscope (DataPhysics Instruments GmbH, Filderstat, Germany). Three measurements were performed, and the average contact angle was estimated using digital images.

### 2.4. Photocatalytic Degradations of Niflumic Acid

Two sets of photocatalytic degradations tests were performed, one with natural and the other with artificial sunlight, with different UV radiation intensities. The photocatalytic degradation of NFA under artificial sunlight radiation was performed using a UV lamp from Phillips (PL-L 24W/10/4P) with a maximum wavelength peak at 365 nm and intensity near 6 W/m^2^. The lamp was kept at 15 cm from the photoreactor containing the NCM and the NFA solution for 6 h. The photocatalytic degradation of NFA with natural sunlight radiation was performed in a solar photoreactor located north of Algeria (latitude 36.39°; longitude 2.42°; sea level). The experiments were performed during the summer season (August 2021), from 10 a.m. to 4 p.m. A Pyranometer CMP 11 (Kipp and Zonen), with a spectral range between 285 and 2800 nm, was used to measure the UV radiation intensity.

The photoreactor, developed at the Solar Equipment Development Unit (UDES) in Algeria, was fabricated from Pyrex glass with a capacity of 1 L (38 × 12 × 8.5 cm), and the TiO_2_/PVDF-TrFE NCM was placed at the bottom of the reactor, as illustrated in Figure 1. A flow rate of 28 mL/s was used to recirculate the NFA solution, so the NCM was always covered with the contaminant solution. Furthermore, the photoreactor was entirely covered with glass to avoid evaporation during the photocatalytic experiments.

For the photocatalytic assays, 1 L of an NFA standard solution was added to the membrane’s photoreactor tank and kept in the dark under stirring for 30 min. Later, the solution and the NCM were irradiated for 6 h, and aliquots were withdrawn at specific time intervals. Different NFA concentrations were tested, ranging from 10 to 30 mg/L, as well as different pH values: 3, 4.7, and 9.

The concentration of NFA was measured by UV–Vis spectrophotometry, using a Shimadzu-1800 spectrophotometer, and by measuring the absorbance intensity at 287 nm, as reported in previous works [36,37] and after observation of the maximum absorption peak. The degradation efficiency and the photocatalytic degradation kinetics were estimated using the Equations (1) and (2), respectively.
(1)$$Degradation\;\left(\%\right)=\frac{C}{C_{0}}\times 100$$
(2)$$n\left(\frac{C_{0}}{C_{t}}\right)=K_{app}t,$$
where *C* represents the degraded NFA concentration, and it corresponds to (*C*_0_–*C_t_*). *C*_0_ and *C_t_* are the NFA initial concentration and the one at a specific reaction time *t* (min), respectively. *K_app_* is the pseudo-first-order rate constant (min^−1^).

High-performance liquid chromatography (HPLC) was employed to assess NFA mineralization, using the following experimental conditions: Walters, USA, with a UV detector at 358 nm; a Diamonsil (R) C_18_ column (5 µm × 150 mm × 4.6 mm ID); mobile phase composed by a combination of distilled water and acetonitrile (90/10, *v*/*v*). The flow rate was set at 0.4 mL/min, the injection volume was 2 μL, and the temperature of the column chamber was kept at 25 °C. A ChemStation software recorded the data.

## 3. Artificial Neural Network

In this study, a three-layered feed-forward neural network with a back-propagation algorithm is considered to predict the photocatalytic degradation efficiency of NFA by the TiO_2_/PVDF-TrFE NCM under solar radiation. The approach followed for determining the optimal network architecture consists of implementing a loop in the MATLAB software to vary the number of the hidden neurons from 1 to 10. Four inputs were considered for the ANN modelling–initial NFA concentration, pH of the media, irradiation time, and radiation intensity (Appendix A). The removal efficiency of the pollutant was selected as the output. Statistical parameters such as a root-mean-squared error (RMSE), mean squared error (MSE), mean absolute error (MAE), and mean absolute percentage error (MAPE), calculated according to previous works [38,39], were used for the evaluation of ANN model accuracy. This study selects the correlation coefficient (R^2^) between the experimental and simulated output as an assessment tool. The network architecture with minimum RMSE, MSE, MAE, and MAPE, and maximum correlation coefficient R^2^ during validation is optimal. The R^2^ between the experimental and simulated output is selected as an assessment tool in this study.

## 4. Results and Discussion

### 4.1. Nanocomposites Characterization

NCMs morphology was evaluated by SEM images (Figure 2a and Appendix A). A well-distributed porous microstructure, along with interconnected spherical pores with an average diameter of ≈60 µm was promoted by the slow evaporation of the solvent [1]. It was found that the incorporation of TiO_2_ NPS slightly affected the morphology of the membrane, as observed by the presence of inner spaces between the pores as a result of possible chemical interactions between NPs and polymeric chains. Additionally, TiO_2_ agglomerates inside the membrane pores and attached to the pore walls were identified by the presence of round white dots (see inset in Figure 2a).

The crystalline structure of TiO_2_ nanoparticles was assessed by XRD (Appendix A). The obtained diffractogram shows the characteristic peaks of anatase (25.2°, 37.8°, 47.9°, and 62.7°) and rutile (27.3°, 54.1°, and 55.2°) crystalline phases of TiO_2_. The diffraction peaks are similar and in agreement with the standard spectra (JCPDS nos: 88-1175 and 84-1286) [40].

FTIR allowed us to identify the polymer phase and possible chemical interactions between photocatalyst and polymer substrate. The FTIR spectra of TiO_2_/PVDF–TrFE NCMs shown in Figure 2b is characterized by the PVDF vibrational modes in the β-phase (840, 1288, and 1400 cm^−1^), characteristic of the co-polymer at the present co-polymer ratio (70/30) [35]. Additionally, the vibration spectra and the polymer phase remain unchanged after TiO_2_ immobilization in the polymeric matrix, indicating that the presence of the NPs does not influence the crystallization phase of the polymer and that no chemical bonds were detected between polymer and photocatalyst, are reported in previous works related to the immobilization of TiO_2_ in PVDF-based substrates [20].

Membranes’ wettability was evaluated through contact angle measurements (Figure 2c). Pristine PVDF-TrFE and TiO_2_/PVDF-TrFE NCMs shown contact angles of 97° and 76°, respectively. These results indicated a vital improvement in terms of wettability, as result of the change from the hydrophobic nature of PVDF-TrFE to hydrophilic nature of NCM, allowing a higher interaction between the photocatalyst and the contaminant. The surface chemistry modifications, as result of inclusion of the nanoparticles, can partially explain these differences. Chemical interactions between TiO_2_ NPs and the polymeric chains result in the formation of inner spaces and consequent formation of pores, as seen by SEM images. The formation of these inner spaces, particularly on the NCM surface, induces a different topography and an increase of hydrophilicity and pore interconnectivity [13].

### 4.2. Photocatalytic Degradation of Niflumic Acid

The evaluation of the photocatalytic efficiency of the TiO_2_/PVDF-TrFE NCMs was performed under different experimental conditions, including the initial NFA concentration (10, 20, and 30 mg/L), pH (3, 7, and 9), irradiation source (solar and simulated), and radiation intensity (552 and 816 W/m^2^).

Two initial experiments were conducted as a control to understand the role of adsorption and photolysis on NFA degradation process (Appendix A). For the adsorption assay, the solution was placed in contact with the NCM without any light source. The NFA solution was placed under solar radiation for the photolysis assay without the NCM. After 6 h of experiment, an adsorption of 18% of NFA was observed, indicating that the NCMs have a slight affinity for NFA. The photolysis experiment showed no degradation after 6 h, confirming the high stability and resilience of NFA and the need for photocatalytic processes [13].

#### 4.2.1. Effect of the Initial Concentration of Niflumic Acid

From a practical point of view, the initial concentration of NFA is one of the main operational parameters affecting degradation efficiency. Therefore, this dependence was evaluated by performing degradation assays using different NFA initial concentrations (Figure 3a).

The results confirmed that the degradation efficiency is strongly dependent on the initial NFA concentration, decreasing from 91% to 59% with increasing concentration from 10 to 30 mg/L. Degradation efficiencies, pseudo-first-order rate constants, and the corresponding R^2^ values are presented in Appendix A, where it is confirmed the decrease of the degradation rate constants from 0.28 to 0.11 min^−1^ with the increase of the NFA concentration from 10 to 30 mg/L.

The degradation efficiency and kinetics are related to the TiO_2_ surface area available for the generation of hydroxyl radicals. The amount of photocatalyst remained constant and, therefore, the amount of hydroxyl radicals generated, while the NFA concentration increased. Thus, the ratio of ${\mathrm{OH}}^{\cdot }$ radicals/NFA molecules decreased for higher concentrations, which lead to a decrease in photocatalytic efficiency [13].

#### 4.2.2. Effect of pH of the Media

The pH of the media is one of the most affecting parameters concerning the photocatalytic efficiency, as it can affect the surface charge of the photocatalyst and contaminant species, which can be determinant for NFA adsorption along the process [41]. The effect of the pH on the NFA degradation efficiency was evaluated in the pH range of 3–9. Figure 3b shows that the removal efficiency of NIF is not significantly affected by the pH of the media. NFA has been reported as positively charged in acidic environments and negatively charged under alkaline conditions, being its point of zero charges (PZC) at a pH of around 5 [42]. Further, TiO_2_ presents a similar surface charge profile, being its PZC at a pH of 6.5 [43]. The similar surface charges of both photocatalyst and contaminant is thus related to the similar degradation efficiencies for the different pH’s under evaluation. The lower degradation kinetics under neutral pH conditions is associated with the lack of surface charge for both photocatalyst and NFA molecules once this pH is close to their PZC, leading to unfavorable conditions for the adsorption of NFA molecules on the NCM surface [44].

#### 4.2.3. Effect of Irradiation Source and Radiation Intensity

The effect of the irradiation source was evaluated under artificial and natural solar radiation. The obtained results are present in Figure 3c. Under solar radiation, a degradation efficiency of 91% was achieved, while under artificial solar radiation, just 33% of NFA was degraded. These differences may be attributed to the range and energy of both artificial and natural solar radiation. Artificial solar radiation presented an intensity of 6 W/m^2^, while natural solar radiation can achieve much higher intensities. Considering that a more energetic UV radiation (short ultraviolet radiation, λ < 300 nm) is present in solar radiation, and that some of the visible radiation have energy higher than bang gap energy of the photocatalyst, under these circumstances TiO_2_ is activated by highly energetic UV and visible radiations and generates a larger amount of hydroxyl radicals, increasing the degradation efficiency [45].

To investigate the effect of weather conditions on the degradation efficiency, the photocatalytic removal was evaluated under solar radiation on a sunny day and cloudy day, presenting different radiation intensities: 816 and 552 W/m^2^, respectively (Figure 3d). It was thus proven that a higher radiation intensity lead to higher degradation efficiency and rate, mainly due to the higher energy of activation provided to TiO_2_ and consequent higher amount of hydroxyl radicals generated [41]. A degradation efficiency decrease of 58% was observed between the experiments performed on sunny and cloudy days.

### 4.3. Reusability of the Nanocomposite Membranes

The reusability of the TiO_2_/PVDF-TrFE NCM was assessed by three consecutive photocatalytic degradation cycles. Between each cycle, the NCMs were washed with UP water and were dried at room temperature. Afterwards, a new NFA solution was placed in contact with the NCM, and the next cycle was performed under the same experimental conditions. The results are presented in Figure 4.

After the first cycle, all the NFA was degraded after 6 h. In the second and third cycles, an efficiency of 91% was obtained, representing an efficiency decrease of 9% when compared to the first cycle. The efficiency loss between the first and second cycles is attributed to the loss of ineffectively attached TiO_2_ NPs on the NCM surface, which are detached during the washing process after the first cycle [46]. As all the efficiently attached NPs remain in the polymeric matrix, no efficiency loss was observed between 2nd and 3rd cycles. The reusability ability of the NCMs is a relevant feature since it allows us to reduce operating costs and environmental impact of the wastewater treatment processes. The results indicate the suitability of the herein prepared TiO_2_/PVDF–TrFE nanocomposite membrane as a promising approach for the long-term degradation of organic pollutants.

### 4.4. Mineralization and Degradation Pathways

The photocatalytic degradation of organic compounds typically results in the decomposition of their chemical structure and may result in the formation of intermediary by-products or in their complete mineralization to H_2_O, CO_2_, and NH_4_^+^. To understand the potential formation of intermediate by-products through the degradation process, the NFA presence was monitored before and after the degradation process through HPLC (Figure 5).

A significant difference was noted by comparing the HPLC chromatograms before and after the degradation process. Besides the substantial decrease of the NFA peak intensity, there was no additional peaks observed, which indicated that, under these conditions, there was no formation of intermediary compounds after 6 h of degradation.

The degradation mechanism of NFA still remains a challenge nowadays. Previous works reported the analysis of by-products of NFA degradation by advanced oxidation processes and the degradation of similar organic compounds using TiO_2_-based photocatalysts [5,47], allows us to propose a degradation mechanism (Figure 6).

According to previous works [5,47], a possible pathway for NFA degradation starts with the adsorption of the NFA molecule on the NCM surface, where further reactions occurred with the hydroxyl radicals generated by the TiO_2_. The first step is the split of both aromatic rings through amine hydrolysis–breaking the N–C bond from the secondary amine and left-aromatic ring, respectively. As a consequence, by-products I and II are formed [5]. Then, by-product I suffers consequent photohydrolysis of –CF_3_ bonds until the formation of by-product III, due to the electron-withdrawing effect of the carboxylic functional group, which hinders the release of an halogen ion unless it is helped by heterolytic scission of H_2_O [47]. In by-product II, the amine group’s C–N bond is broken to generate by-product IV. Then, both by-products III and IV continue to react with ${\mathrm{HO}}^{\cdot }$ radicals, resulting in the phenolic by-products V and VI [48]. Finally, the opening of aromatic rings happens (VII and VIII), and the complete mineralization of all by-products into CO_2_, H_2_O, and NH_4_^+^ is achieved.

### 4.5. Artificial Neural Network Modeling Results

An ANN model was developed to investigate the predictability of NFA degradation efficiency. Data results were obtained by changing the different experimental conditions and dividing them into input and output matrices. The input layer contained the four operational variables: initial NFA concentration, pH of the media, irradiation time, and radiation intensity. The output layer consisted of only one neuron, the NFA degradation efficiency. After data scaling, a random splitting of the dataset was performed into learning (60%), validation (20%) and test (20%) sets.

The plot of the variation for the RMSE for both training, validation, and test sets, as a function of the number of the hidden neurons is provided in Figure 7a. The accumulated RMSE values close to the y = 0 line indicate the suitability of the ANN model. For a suitable model structure, all the RMSE values must be positioned between −1 and 1 [45]. As observed, the network topology with eight hidden neurons layer achieved the minimal validation RMSE (0.0031) and, therefore, it represented the best perdition result among the 10 networks generated by the *MATLAB* program.

To assess the accuracy of the optimal model (4:8:1), which contained four input neurons, eight hidden neurons and one output neuron, the predicted output values were compared to the corresponding experimental values (Figure 7b). The R^2^ for training, validation and test sets were 0.98, 0.997, and 0.96, respectively. The accumulated values close to the *y* = *x* regression indicate the suitability of the ANN model to predict NFA degradation, as they show a strong linear relationship between experimental and predicted data. So, the 4:8:1 artificial neural network model (Figure 7c) provides an effective tool to simulate the non-linear behavior of the photocatalytic degradation of niflumic acid by the 10 wt.% TiO_2_/PVDF-TrFE nanocomposite membranes, and it is highly recommended to predict the NFA degradation efficiency.

Statistical parameters were required to understand further the suitability of the ANN model to predict NFA degradation by NCMs. In this scope, R^2^, RMSE, MAE, and MAPE were used to evaluate the accuracy of the ANN model (Table 1). The low RMSE, MAE, and MAPE values show a good fit for the ANN model, and, in addition, the R^2^ value close to 1 indicates an excellent correlation between experimental and predicted results. These results demonstrate that ANN modelling can forecast the performance for NFA degradation, considering irradiation time, media pH, radiation intensity, and initial NFA concentration.

ANN modelling has been increasingly applied in many fields as a fundamental tool to resolve forecasting challenges. Nonetheless, these empirical models are usually considered unable to clarify the contribution of the independent variables to the dependent ones. Taking this into account, extensive research on the analysis of the relative impact of the input variables on the neural network response to make ANNs more interpretable has been made [49]. The relative relevance values of the different input variables were evaluated using Garson’s algorithm [50]. According to this algorithm, the input–hidden and the hidden–output weights of the model are subdivided. The absolute values of the weights are used to calculate the relative relevance of the input variables on the output response. The available data are divided into two subsets: (1) training and validation data; and (2) testing data. The weights matrix of the optimal neural network is given in Appendix A. The main purpose of applying the Garson’s algorithm was to understand the relevance of each experimental parameter and their contribution to the prediction of the NFA degradation efficiency by the NCM. The results are summarized in the Table 2.

From Table 2, it is possible to observe that the input variables have a substantial effect on NFA degradation efficiency. The irradiation time is considered the most critical parameter affecting the NFA degradation, with a relative relevance of 33.4%. The pH of the media (25.7%) and radiation intensity (22.7%) have a significant impact on NFA degradation, while initial NFA concentration (18.2%) proves to be the less relevant parameter. These results agree with previous works reporting the use of TiO_2_-based photocatalysts for the photocatalytic degradation of pharmaceuticals, as it is essential for the generation of hydroxyl radicals, triggering further reactions and the cleavage of compound chemical bonds [41]. This deep understanding of the relative relevance of experimental parameters represents a critical step for effectively implementing water remediation systems based on photocatalytic reactors to remove contaminants of emerging concern from water sources.

## 5. Conclusions

Developing new strategies for the efficient photocatalytic degradation of persistent and hazardous contaminants from water is a pressing concern. The present work reported the development of 10 wt.% TiO_2_/PVDF–TrFE nanocomposite membranes by solvent casting and their implementation on a solar photocatalytic reactor for the degradation of NFA.

The prepared NCMs present a homogeneous micrometric porous structure and a homogeneous distribution of the TiO_2_ nanoparticles within the interconnected pores. The photocatalytic performance of the NCM was evaluated through the degradation of NFA under solar irradiation and different experimental conditions, such as the initial NFA concentration, pH of the media, irradiation time and radiation intensity. Irradiation time was the most significant parameter affecting the performance of the NCM, and an equilibrium was achieved after 6h of irradiation. Initial NFA concentration proved to be a predominant parameter affecting the photocatalytic efficiency, and a maximum efficiency was achieved in the presence of a 10 mg/L NFA solution (91%), which decreased by increasing the NFA concentration (59% for a 30 mg/L NFA concentration). The pH of the media did not significantly affect the performance of the system, being relevant point out that the maximum performance was obtained under a neutral environment (91%). The NCM proved to be more effective when irradiated by solar light, achieving a significantly higher efficiency when compared by artificial solar radiation (91 vs. 33%). Also, the solar radiation intensity and the effect of weather conditions was evaluated, and an efficiency decrease of 58% was noted by decreasing the radiation intensity from 816 to 552 W/m^2^. The reusability of the NCM after three consecutive uses caused an efficiency loss of 9%. Further, the system was theoretically evaluated by an ANN model. The ANN model was successfully implemented to predict the photocatalytic process, as a good fit was achieved for the comparison between experimental and predicted results, as well as the high R^2^ (0.98) and reduced RMSE values (0.013). In addition, the ANN model allowed us to estimate the relative relevance of the experimental parameters, making the irradiation time the most affecting parameter for the NFA degradation efficiency (33.4%).

Thus, the 10 wt.% TiO_2_/PVDF–TrFE nanocomposite membranes and its implementation on a solar photocatalytic reactor represent a suitable up-scalable technological approach for the degradation of niflumic acid and related pollutants from water under solar irradiation.

## Figures and Tables

**Figure 1 membranes-12-00849-f001:**
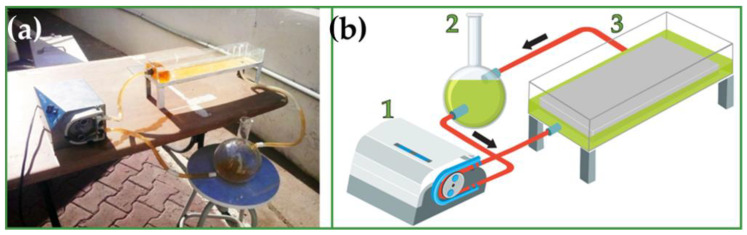
(**a**) Real image and (**b**) schematic representation of the developed solar photoreactor.

**Figure 2 membranes-12-00849-f002:**
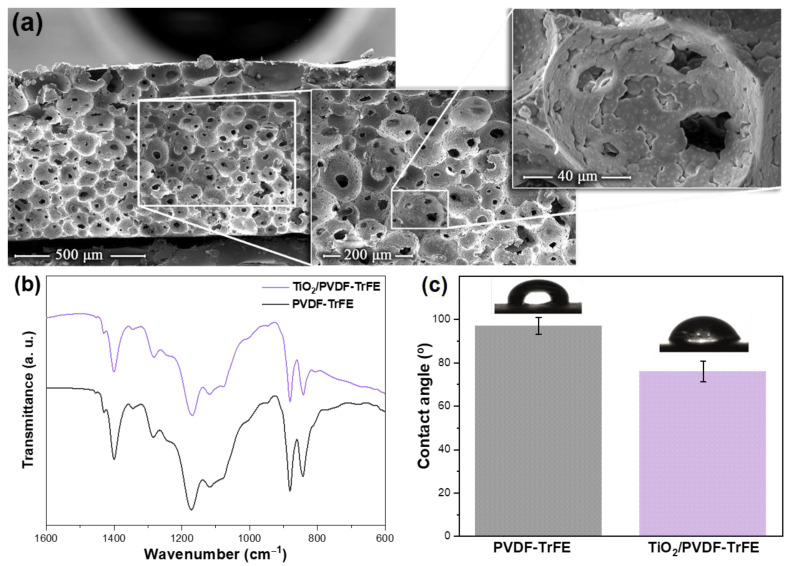
(**a**) Representative SEM cross-section images of TiO_2_/PVDF–TrFE NCMs with different magnifications; (**b**) FTIR spectra and (**c**) contact angle of PVDF-TrFE membranes and TiO_2_/PVDF–TrFE NCMs.

**Figure 3 membranes-12-00849-f003:**
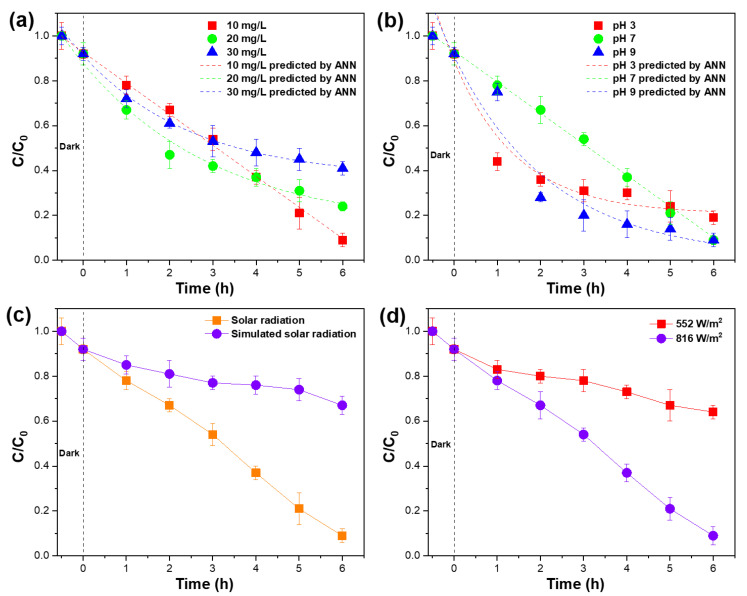
Effect of (**a**) initial NFA concentration (recirculation time: 6 h; pH = 7), (**b**) pH ([NFA] = 10 mg/L; recirculation time: 6 h), (**c**) irradiation source, and (**d**) radiation intensity on the photocatalytic degradation of niflumic acid ([NFA] = 10 mg/L; recirculation time: 6 h; pH = 7).

**Figure 4 membranes-12-00849-f004:**
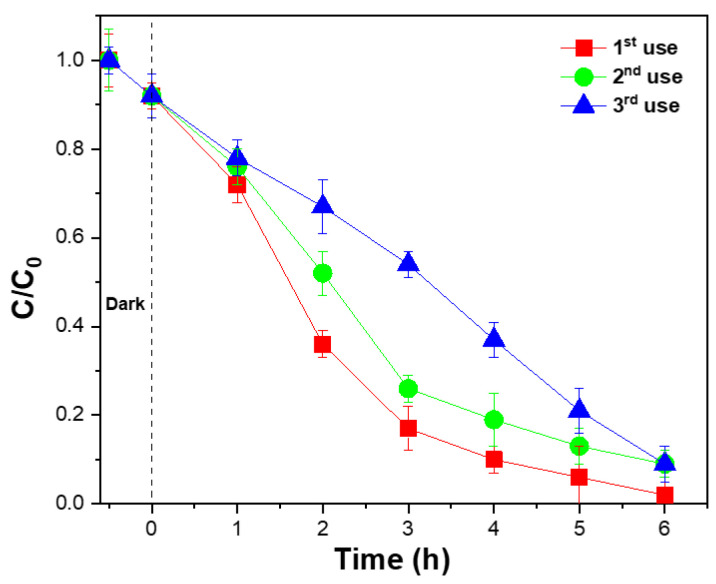
Photocatalytic degradation of NFA with the TiO_2_/PVDF–TrFE nanocomposite membranes in three consecutive uses, under solar irradiation ([NFA] = 10 mg/L; recirculation time: 6 h; pH = 7).

**Figure 5 membranes-12-00849-f005:**
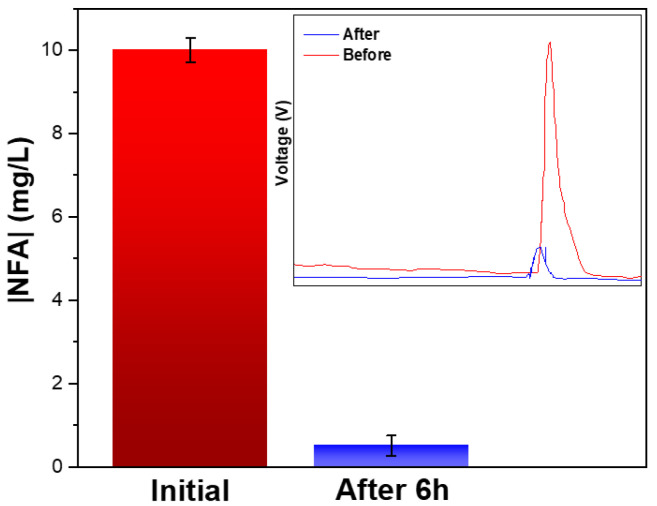
Photocatalytic degradation of NFA with the TiO_2_/PVDF–TrFE nanocomposite membranes under solar irradiation. Inset: HPLC chromatogram of NFA samples before and after 6 h of degradation ([NFA] = 10 mg/L; recirculation time: 6 h; pH = 7).

**Figure 6 membranes-12-00849-f006:**
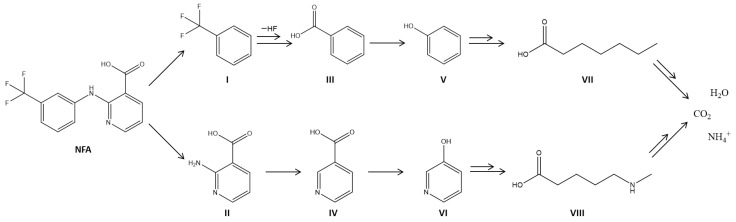
Schematic representation of the proposed photocatalytic degradation mechanism NFA (adapted from [5,47]).

**Figure 7 membranes-12-00849-f007:**
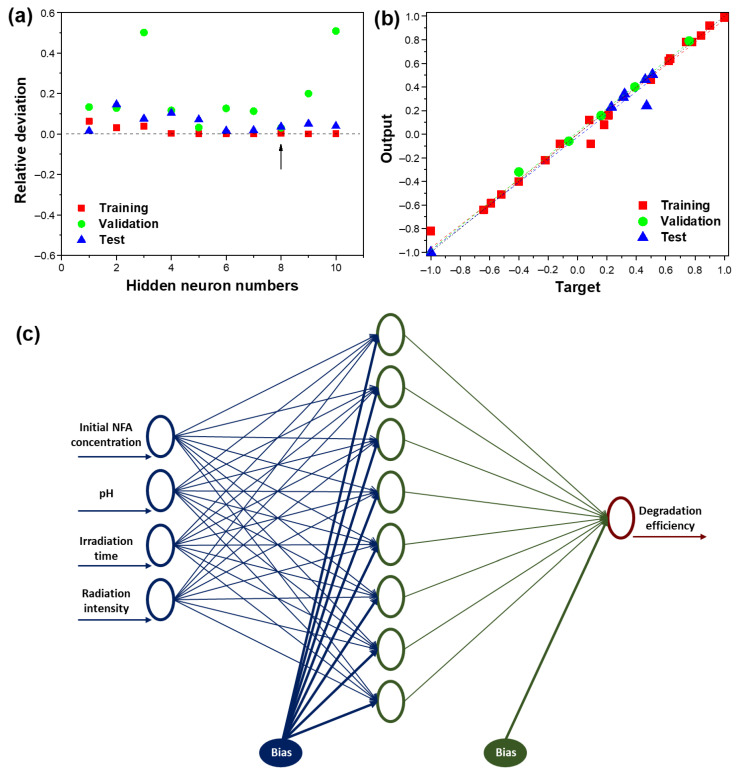
(**a**) Variation of RMSE for training, validation, and test sets, as function of hidden neurons number; (**b**) experimental results versus predicted ones for training, validation, and test sets; (**c**) structure of the optimal neural network (4:8:1).

**Table 1 membranes-12-00849-t001:** Statistical parameters of the developed ANN model.

Statistical Parameters	Value
R^2^	0.98
RMSE	0.013
MAE	0.020
MAPE	0.079

**Table 2 membranes-12-00849-t002:** Relative relevance of the process input variables.

Input Variable	Relative Relevance (%)	Rank
Initial NFA concentration (mg/L)	18.2	4
Initial pH	25.7	2
Irradiation time (h)	33.4	1
Solar irradiation intensity (W/m^2^)	22.7	3

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
