# Peer review of "Solar Photocatalytic Membranes: An Experimental and Artificial Neural Network Modeling Approach for Niflumic Acid Degradation"

_membranes, 2022, doi:10.3390/membranes12090849_

Round 1

Reviewer 1 Report

Reviewer comments_Membranes 1873260

The authors of this manuscript Nanocomposite membranes for solar photocatalytic degradation of niflumic acid: a combined experimental and artificial neural network modeling approach. The findings reported in this manuscript deserve to be known by other researchers. However, this manuscript requires major revision and before publication, several questions should be illustrated more clearly to make the manuscript more readable and meaningful to readers. Detailed comments are as follows:

  1. Kindly include the exact time (pm or am) and the type of season when the photocatalytic measurement was conducted using natural sunlight.
  2. Include the reason why the added NFA standard solution in the membrane photoreactor tank was kept in the dark under stirring for 30 min before light irradiation? What is the reason for this?
  3. Kindly confirm and add references of NFA wavelength absorbance measured around this value (287 nm).
  4. Any specific reason why 10wt% of TiO2 was chosen in the formulation?
  5. Kindly elaborate further on this statement “These results indicated a change from the hydrophobic nature of PVDF-TrFE to hydrophilic nature of NCM due to surface chemistry modifications as a result of the inclusion of the nanoparticles, allowing a higher interaction between the photocatalyst and the contaminant [13].“ How do the wettability properties of PVDF-TrFE upon the addition of TiO2?
  6. Kindly add SEM crosssection images of both PVDF-TrFE membrane and TiO2/PVDF-TrFE membranes and compare their morphology and discuss how they relate with the wettability and membrane performance
  7. Title needs to be revised and specific to the project done
  8. I would suggest the authors include the filtration performance (pure water flux and NFA permeate flux) in this manuscript as the authors highlighted the materials as membranes, not as photocatalytic materials

Author Response

We are grateful for the reviewer's recognition of our effort, as well as the suggestions for the improvement of this work.

Reviewer 2 Report

The paper is well presented and well described. The results portrayed extensive experimental design to fulfill the objectives of the research. Just a few comments to improve the paper:

1.       Line 177-210

The justification of using ANN can be placed in the introduction. In the methodology, just write the protocol used in the study.

2.       Equation 1 – Indicates whether this equation refers to C/Co. Otherwise, please include the equation for C/Co.

3.       Line 297 – Considering that a more energetic UV radiation (short ultraviolet radiation, λ < 300 nm) is present in solar radiation, and that some of the visible radiation have energy higher than bang gap energy of the photocatalyst, under these circumstances TiO2 is activated by highly energetic UV and visible radiations and generates a larger amount of hydroxyl radicals, increasing the degradation efficiency [44].

Band gap

4.       Figure 5. The concentration of NFA degraded is best presented in concentration.

5.       Conclude the manuscript with all the results obtained from the experiments.

Author Response

We are grateful for the reviewer's compliments and recognition of our effort on this work.
